# Comparing T cell receptor repertoires using optimal transport

**Branden J. Olson**[1,2], **Stefan A. Schattgen**[3], **Paul G. Thomas**[3], **Philip Bradley**[1,4]*, **Frederick A. Matsen IV**[1,2,5,6]*

**1** Department of Computational Biology, Fred Hutchinson Cancer Research Center, Seattle, Washington, United States of America, **2** Department of Statistics, University of Washington, Seattle, Washington, United States of America, **3** Department of Immunology, St. Jude Children's Research Hospital, Memphis, Tennessee, United States of America, **4** Institute for Protein Design, Department of Biochemistry, University of Washington, Seattle, Washington, United States of America, **5** Department of Genome Sciences, University of Washington, Seattle, Washington, United States of America, **6** Howard Hughes Medical Institute, Seattle, Washington, United States of America

* pbradley@fredhutch.org (PB); matsen@fredhutch.org (FAM)

## Abstract

The complexity of entire T cell receptor (TCR) repertoires makes their comparison a difficult but important task. Current methods of TCR repertoire comparison can incur a high loss of distributional information by considering overly simplistic sequence- or repertoire-level characteristics. Optimal transport methods form a suitable approach for such comparison given some distance or metric between values in the sample space, with appealing theoretical and computational properties. In this paper we introduce a nonparametric approach to comparing empirical TCR repertoires that applies the Sinkhorn distance, a fast, contemporary optimal transport method, and a recently-created distance between TCRs called TCRdist. We show that our methods identify meaningful differences between samples from distinct TCR distributions for several case studies, and compete with more complicated methods despite minimal modeling assumptions and a simpler pipeline.

**Data Availability Statement:** All relevant data are within the manuscript and corresponding GitHub repository https://github.com/matsengrp/transport.

## Author summary

T cells are critical for a successful adaptive immune response, largely due to the expression of highly diverse receptor proteins on their surfaces. These T cell receptors (TCRs) recognize peptides that may be foreign invaders such as viruses or bacteria. Because of this, immunologists are often interested in comparing different sets (or repertoires) of these TCRs in hopes of identifying groups of particular interest, such as TCRs that are responding to a particular vaccination using pre- and post-vaccination samples. Current methods of comparing TCR repertoires either rely on statistical models which may not adequately describe the data, use summary statistics that may lose information, or are difficult to interpret. We present a complementary method of comparing TCR repertoires that detects significantly different TCRs between two given repertoires using a distance rather than a model, summary statistics, or dimension reduction. We demonstrate that our

**Funding:** BJO received funding from the National Institutes of Health (https://www.nih.gov/) R01 AI146028, U19 AI117891. SAS and PGT received funding from the National Institutes of Health (https://www.nih.gov/) R01 AI121832, R01 AI136514, U01 AI150747, and the American Lebanese Syrian Associated Charities at St. Jude (https://www.stjude.org/about-st-jude/faq/whats-alsac.html). PB received funding from the National Institutes of Health (https://www.nih.gov/) R35 GM141457, R01 AI146028. FAM received funding from the National Institutes of Health (https://www.nih.gov/) R01 AI146028, U19 AI117891, a Faculty Scholar grant from the Howard Hughes Medical Institute (https://hhmi.org/) and the Simons Foundation (https://www.simonsfoundation.org/). The funders had no role in study design, data collection and analysis, decision to publish, or preparation of the manuscript.

**Competing interests:** I have read the journal's policy and the authors of this manuscript have the following competing interests: Paul G Thomas consults for Johnson and Johnson, Immunoscape, Cytoagents, and PACT Pharma. He has received travel reimbursement from 10X Genomics and Illumina. He has patents on methods related to T cell receptor biology. The other authors declare that no competing interests exist.

method can identify biologically meaningful repertoire differences using several case studies.

## Introduction

T cell receptors (TCRs) are protein complexes found on the surfaces of T cells, important white blood cells to the adaptive immune response. Through the ability of their TCRs to bind to foreign invaders like viruses or bacteria, T cells are able to recognize and neutralize these invasions, ultimately allowing for robust and long-lasting immunological protection. The DNA sequences coding for TCRs arise by a complex stochastic recombination process called V(D)J recombination, which includes insertions and deletions in a region known as the complementarity determining region 3, or CDR3. Even after a series of productivity-based filters ensuring functionality and limiting self-reactivity [1], this process yields an astronomical diversity in the circulating pool of TCR sequences. Thus, proper analysis of TCR repertoires as well as the immune context surrounding them presents a formidable but necessary challenge to computational biologists.

The arrival of high-throughput sequencing has given scientists the ability to sample TCR repertoires with unprecedented depth, paving the way for immense progress within the field of computational immunology. Often, this reduces to a situation wherein a researcher wishes to compare two TCR repertoire datasets and extract meaningful differences between them. For example, the pair of datasets could be samples of an individual's TCR repertoire before and after a vaccination, and the researcher might wish to determine the responding TCRs in the post-vaccination repertoire.

Most current methods of repertoire comparison involve reducing the TCR sequences into simpler summaries and then comparing these summaries [2–10]. As comparing full CDR3 sequences can be highly involved, one approach is to simply compare CDR3 length distributions [11, 12]. These approaches fail to capture other interesting aspects of the germline-encoded regions such as gene similarity, as well as the relative importance of the CDRs and framework regions for TCR binding specificity. Alternatively, one can project a TCR repertoire onto a simpler space and compare values within the resultant embedding. For example, several studies have examined the distributions of *k*mer occurrences to classify TCR repertoires [13–15]. However, the space of *k*mer distributions is still very high-dimensional and discards important positional information within TCR sequences. Other approaches instead look at t-SNE projections of repertoires [16], but this still incurs a loss of information and loses immunological meaning. Yet another line of work analyzes clone frequency between samples, without a means of assessing sequence similarity between the clones [17–19].

The more focused problem of inferring specificity from one or more samples of TCR sequence has been approached by tracking individual clones or comparing to a probabilistic model, as follows. Tracking distributions of clonotypes over multiple timepoints [20] typically requires at least three longitudinal datasets per individual. A recent technique addresses this issue by detecting TCRs within a single repertoire that are significantly enriched according to some baseline generative model [21]. While this is a substantial advance, the method depends on the underlying generative model being accurate.

Another line of work uses experimentally-inferred antigen-associated TCRs as labeled data; this is a different setting than the one approached in this paper. For example, machine learning techniques can be used to build predictive models using these labeled training data [22–24]. These can be limited by the amount of publicly-available data, and rely on models that can be

difficult to interpret. Another approach is to cluster sequences based on amino acid similarity with the goal of grouping TCRs that respond to the same epitope [25–28].

We wished to develop a procedure that performs comparisons between two empirical repertoires in a fast, interpretable, and precise manner. Thinking of a sample TCR repertoire as an empirical distribution of observed sequences, the problem reduces to comparing two discrete probability distributions using some measure of statistical divergence. There are many methods for comparing discrete distributions, but many of these methods are hardly appropriate for TCR datasets, which comprise a very sparse sample from the very vast space of possible TCRs. One way to assuage this sparsity is to equip the sample space of TCR sequences with some metric which provides distributional comparison. While several such metrics on probability distributions have been established, we focus on a particular class of methods known as optimal transport, which boasts favorable theoretical and computational properties along with an intuitive interpretation. Moreover, while classical optimal transport methods are computationally intensive and often scale at least cubically with the number of statistical parameters, a recent extension uses "Sinkhorn distances" that regularize the underlying optimization problem to get tractable approximations with high accuracy [29].

In this report we apply the Sinkhorn approach along with TCRdist, a recently-developed distance between TCRs [30], to formulate a nonparametric approach to the comparison of TCR repertoires. We motivate our methods using the intuition underlying optimal transport, and demonstrate that our methods are able to identify clusters of TCR sequences that constitute biologically meaningful differences between repertoires using multiple case studies. We also describe and validate a randomization test to assess whether our identified TCRs are significantly enriched in a target repertoire with reference to a source repertoire.

## Materials and methods

### An optimal transport formulation of TCR repertoire comparison

Optimal transport compares two probability distributions in terms of the total amount of "work" required to transform one probability distribution into the other. In this context, work is defined as the product of the probability mass (i.e., the normalized frequency of a value's occurrence) between objects in the joint sample space and the distance between them (according to some specified distance function). To illustrate, one might think of these distributions as soldiers on a battlefield: one can compare two distributions of soldiers by the minimal amount of overall work (total amount of marching among all soldiers) that is required to move them from one configuration to another [31]. In this analogy, each unit of probabilistic "mass" corresponds to a single distinct soldier, and higher probabilistic mass at a given location corresponds to a larger number of soldiers at that location. The strength of the optimal transport approach for this application is that it quantifies not only the minimal total amount of transport needed given a particular mapping of mass in one distribution to the other, it also returns a description of how the transport is performed. In our soldier example, this would be the particular marching orders for each soldier concerning how they should be dispatched into the second configuration.

Returning now to TCRs, we can consider each TCR to be a soldier. The "marching distance" is defined by TCRdist [30], so that the result of an optimal transport analysis of two repertoires is a mapping of TCRs in one repertoire to another in which similar TCRs are matched to one another. (Note that we can match part of one TCR to another TCR by assigning a fraction of the probability mass between them, so there is no difficulty in having repertoires of different sizes.) TCRs that have no close relative in another repertoire must travel a long distance, which can be easily identified from the optimal mapping. That is, large values of the optimal

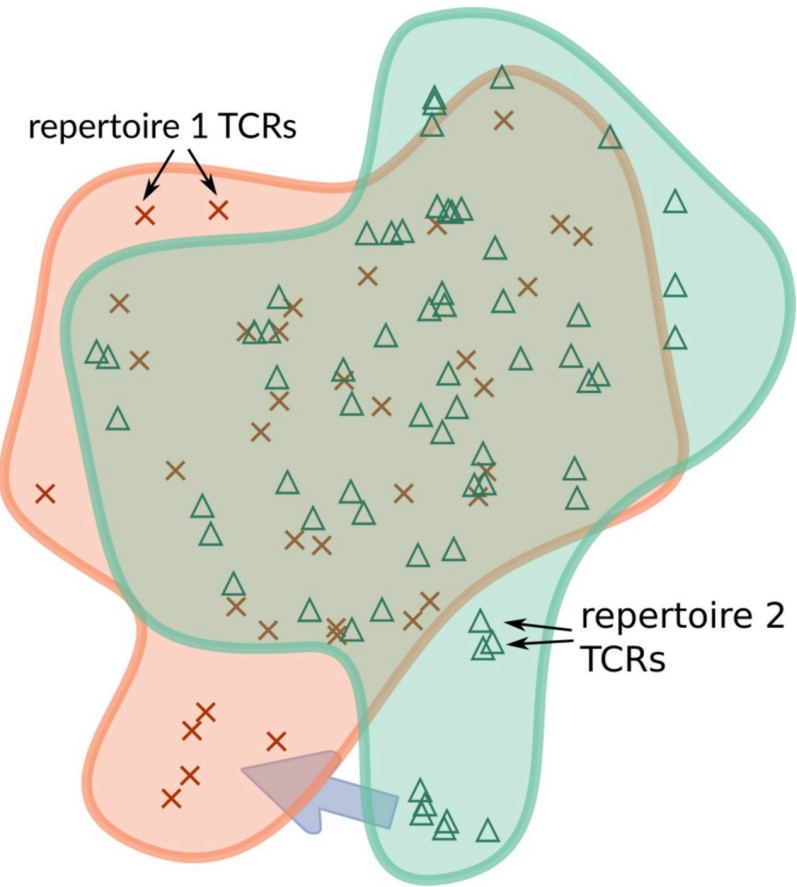

**Fig 1. A schematic of TCR distribution comparison.** Each symbol represents a TCR in an abstract space in which distance is defined via TCRdist [30], and the two regions represent two population repertoires of interest. Each repertoire is given its own color (here orange and green). The purple arrow shows that there are regions of these TCR distributions for the green repertoire that do not have a close equivalent in the orange repertoire, which will be identified by our optimal transport methods.

transport matrix hint at some discrepancy between the underlying TCR distributions. This will in turn allow us to identify regions of "TCR space" that are differentially occupied between the two repertoires (Fig 1). This concept will underlay our methodology to detect notable regional differences between two TCR repertoires.

In the remainder of this section, we briefly review discrete optimal transport and how it can be leveraged to compare TCR repertoires. We then derive a score to detect which individual TCRs appear to be enriched in a target repertoire with respect to a source repertoire using the optimal transport matrix. Using these scores, we develop a procedure to extract clusters of differentially-expressed TCRs, as well as a procedure to infer sequence motifs that characterize these clusters. We also describe a randomization test to obtain statistical significance estimates for these scores. We conclude this section with a discussion of the repertoire datasets that are analyzed in the Results section.

## Discrete optimal transport

Suppose we have two discrete probability distributions described by vectors $\mathbf{r} = (r_1, \ldots, r_n)^\top$, the probability masses assigned to objects $x_1, \ldots, x_n$, respectively, and $\mathbf{c} = (c_1, \ldots, c_m)^\top$, the

probability masses assigned to objects $y_1, \ldots, y_m$, respectively, so that $\mathbf{r}$ and $\mathbf{c}$ contain nonnegative entries, and both sum to one. We can consider the set of *admissible couplings* [32], encoded as joint probability matrices whose row-sums correspond to $\mathbf{r}$ and whose column-sums correspond to $\mathbf{c}$:

$$\mathbf{U}(\mathbf{r}, \mathbf{c}) \coloneqq \{\mathbf{P} \in \mathbb{R}_+^{n \times m} : \mathbf{P}\mathbf{1}_m = \mathbf{r} \ \& \ \mathbf{P}^\mathsf{T}\mathbf{1}_n = \mathbf{c}\}, \tag{1}$$

where $\mathbf{1}_k = (1, \ldots, 1)^\mathsf{T} \in \mathbb{R}^k$. In other words, we are considering all joint probability distributions whose marginal distributions correspond to $\mathbf{r}$ and $\mathbf{c}$. For a given matrix $\mathbf{P}$, we can interpret the entry $p_{ij}$ as the amount of mass "assigned" or "transported" between the object $x_i$ (which has $r_i$ total mass) and object $y_j$ (which has $c_j$ total mass).

By formalizing this in the language of measure theory we will be able to provide a rigorous specification of our methods below. Let $\Sigma_k = \left\{\mathbf{a} \in [0, 1]^k : \sum_{i=1}^k a_i = 1\right\}$ be the standard $k$-simplex. Let $\delta_x(\cdot)$ denote the Dirac delta measure centered on a point $x$, which evaluates to 1 if the input is $x$ and 0 otherwise [33]. Consider discrete probability measures $\mu(\cdot) = \sum_{i=1}^n r_i \delta_{x_i}(\cdot)$ and $v(\cdot) = \sum_{j=1}^m c_j \delta_{y_j}(\cdot)$ on respective sample spaces $\mathcal{X}$ and $\mathcal{Y}$, with $\{x_1, \ldots, x_n\} \subset \mathcal{X}$, $\{y_1, \ldots, y_m\} \subset \mathcal{Y}$, where $\mathbf{r} = (r_1, \ldots, r_n)^\mathsf{T} \in \Sigma_n$ and $\mathbf{c} = (c_1, \ldots, c_m)^\mathsf{T} \in \Sigma_m$ denote the same vectors as above.

For a given distance matrix $\mathbf{D} \in \mathbb{R}^{n \times m}$, such that $\mathbf{D}_{ij}$ is the distance between $x_i$ and $y_j$, the classical "Kantorovich" optimal transport problem seeks the solution of

$$L_\mathbf{D}(\mathbf{r}, \mathbf{c}) \coloneqq \min_{\mathbf{P} \in \mathbf{U}(\mathbf{r}, \mathbf{c})} \langle \mathbf{D}, \mathbf{P} \rangle \tag{2}$$

where $\langle \mathbf{A}, \mathbf{B} \rangle \coloneqq \sum_{i=1}^n \sum_{j=1}^m a_{ij} b_{ij}$ for $\mathbf{A}, \mathbf{B} \in \mathbb{R}^{n \times m}$. That is, we compute the optimal transport matrix $\mathbf{P}$ which minimizes the sum of entrywise products of distance and probability mass. We interpret this as the total amount of "work" to move the mass of one distribution to another. Hence, this distance between probability distributions is often referred to as the Earth-mover's distance (EMD), and is also known as the Wasserstein metric. It is important to note that we are working with two notions of distance: the distance defined on the sample space between two objects and represented by the matrix $\mathbf{D}$, and the overall distance between the two full probability distributions defined by the EMD.

Unfortunately, computing the EMD of two discrete distributions scales as $\mathcal{O}(k^3 \log(k))$, where $k = \max(m, n)$, when no restrictions are placed on the metric $d$ that parametrizes the distance matrix $\mathbf{D}$. Cuturi (2013) overcomes this by regularizing the entropy of the couplings $\mathbf{P}$ which drive the minimization [29]. In particular, they introduce the Sinkhorn distance

$$d_\mathbf{D}^\lambda(\mathbf{r}, \mathbf{c}) \coloneqq \langle \mathbf{D}, \mathbf{P}^\lambda \rangle \tag{3}$$

where

$$\mathbf{P}^\lambda = \arg\min_{\mathbf{P} \in \mathbf{U}(\mathbf{r}, \mathbf{c})} \left\{ \langle \mathbf{D}, \mathbf{P} \rangle - \frac{1}{\lambda} h(\mathbf{P}) \right\} \tag{4}$$

and $h(\mathbf{P}) \coloneqq -\sum_{i=1}^d \sum_{j=1}^d p_{i,j} \log(p_{i,j})$ is the Shannon entropy of $\mathbf{P}$. This regularization serves two main purposes. First, we can interpret the regularization term as an invocation of the principle of maximum entropy, which encodes the intuition that we should choose a distribution with the fewest assumptions (i.e., the most entropy) when considering a set of viable candidate distributions. In addition, the regularization introduces smoothing into the transport plan between $\mathbf{r}$ and $\mathbf{c}$ which leads to an approximate but much faster solution (the tuning parameter $\lambda$ controls this speed-accuracy tradeoff). Cuturi (2013) shows that the regularization term

constrains the optimization region of admissible couplings $\mathbf{U}$ to a new region $\mathbf{U}_\alpha$ such that

$$\mathbf{U}_\alpha(\mathbf{r}, \mathbf{c}) = \{\mathbf{P} \in \mathbf{U}(\mathbf{r}, \mathbf{c}) : \mathrm{KL}(\mathbf{P}||\mathbf{rc}^\mathsf{T}) \leq \alpha\}, \tag{5}$$

where KL denotes Kullback-Leibler divergence. We recall this derivation in Appendix A. Thus, we can interpret the Sinkhorn distance as the result of minimizing the work to move one distribution to another while maintaining a relatively simple coupling, in the sense that its KL-divergence to the independent joint distribution (whose coupling is exactly $\mathbf{rc}^\mathsf{T}$) is small.

## Applying optimal transport to TCR repertoire comparison

For our purposes, the sample spaces $\mathcal{X}$ and $\mathcal{Y}$ discussed above will denote the same set of possible TCR$\beta$ sequences we can observe in a sample TCR$\beta$ repertoire. We can use any encoding of those sequences that is compatible with a distance on TCR$\beta$ sequences. Because we are using TCRdist, we will use an encoding that corresponds with its input: we define this set $\mathcal{X}$ to be all valid pairs $t$ of TRBV genes and CDR3 amino acid sequences, e.g., $t$ = (TRBV27*01, CASSLGTGQYEQYF). We use "empirical repertoire", or simply "repertoire", to mean a repertoire sample $R = (t_1, \ldots, t_n)$ containing $n$ (TRBV, CDR3aa) pairs along with corresponding abundances $(a_1, \ldots, a_n) \in (\mathbb{Z}^+)^n$. We elected to exclude the J gene identity from our analysis, as the J gene's impact on TCR binding/specificity is primarily transmitted through sequence contained in the CDR3 region. Relating this to the notation of the previous section, we have $x_i = t_i$ as the sample points, and $c_i = a_i/\Sigma_k a_k$ as the corresponding mass coefficients; analogous quantities are used to define $y_i$ and $r_i$ for a second repertoire.

For our distance function $d$, we use a version of TCRdist, a similarity-weighted mismatch distance between potential pMHC-contacting loops of two given TCRs [30] that can be applied to paired or single-chain TCR sequence data. Here we focus on single-chain TCR$\beta$ repertoire data, however our approaches extend naturally to comparisons of paired TCR repertoires. If $\mathbf{a}_1^c$ and $\mathbf{a}_2^c$ are the amino acid sequences of CDR $c$ for TCRs 1 and 2 respectively, then

$$\mathrm{TCRdist}(t_1, t_2) \coloneqq \sum_{c \in \mathrm{CDRs}} \sum_{i \in p} w(c)\, \mathrm{AAdist}\left((\mathbf{a}_1^c)_i, (\mathbf{a}_2^c)_i; c\right) \tag{6}$$

where:

- CDRs $\coloneqq$ {CDR1$\beta$, CDR2$\beta$, CDR2.5$\beta$, CDR3$\beta$}

- $w(c) \coloneqq \begin{cases} 3, & c = \mathrm{CDR3}\beta \\ 1, & \text{else} \end{cases}$

- $\mathrm{AAdist}(a_1, a_2; c) \coloneqq \begin{cases} 0, & a_1 = a_2 \\ 4, & \text{exactly one of } a_1 \text{ or } a_2 \text{ is `}-\text{'} \\ \min(4, 4 - \mathrm{BLOSUM62}(a_1, a_2)), & \text{else} \end{cases}$

- BLOSUM62 is a widely-used substitution matrix for amino acids that was estimated using log odds scoring of frequencies from a large alignment database (called BLOCKS) [34].

These formulas are justified in the methods section of the paper defining TCRdist [30].

Fig 2 illustrates a simple example of this setup with two TCR repertoires spread out in an abstract space, where the distance between TCRs is defined by TCRdist. The three orange TCRs spanning in the upper-right of the image belong to some repertoire $R_1$, and the four purple TCRs spanning the bottom of the image belong to some other repertoire $R_2$. Their respective abundances are displayed in the adjacent circles. The dotted lines between TCRs represent

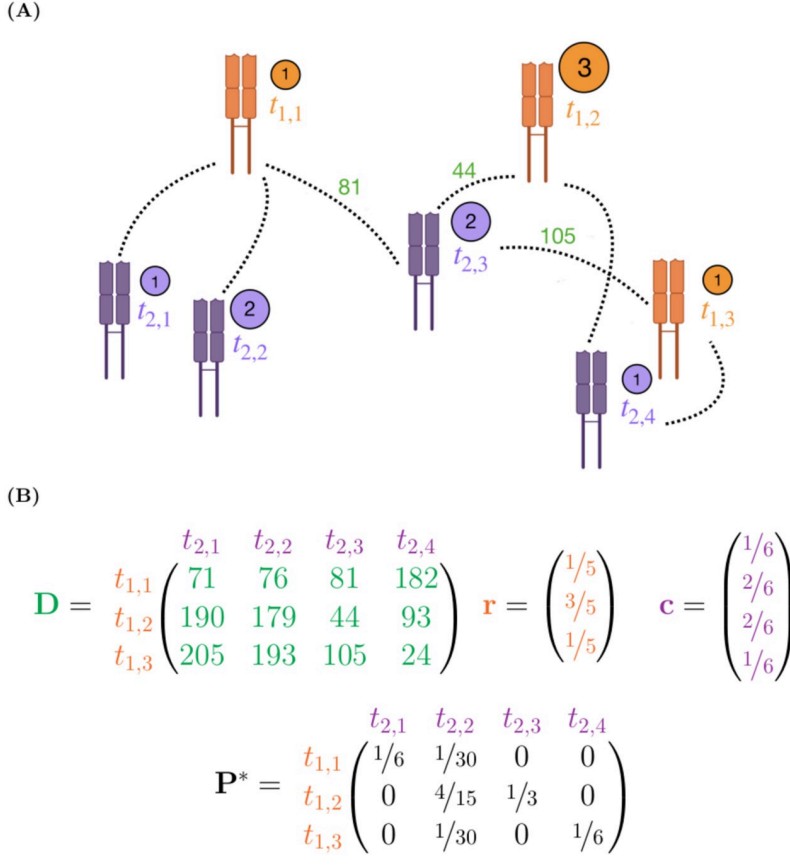

**Fig 2. An illustration of our optimal transport formulation of TCR repertoire comparison. (A)** A schematic of two TCR repertoires $R_1 = \{t_{1,1}, t_{1,2}, t_{1,3}\}$ and $R_2 = \{t_{2,1}, t_{2,2}, t_{2,3}, t_{2,4}\}$ residing in an abstract space defined by TCRdist. The circle adjacent to each TCR displays its clonotype abundance. TCRdist values are shown (in green) from $t_{2,3}$ to each of the TCRs in $R_1$, although a TCRdist value is defined between each pair. **(B)** The mathematical objects that describe the setup illustrated in (A). Here, $\mathbf{D}$ is the matrix of pairwise TCRdist values, $\mathbf{r}$ is a vector of distribution mass values for each TCR in $R_1$, $\mathbf{c}$ is a vector of distribution mass values for each TCR in $R_2$, and $\mathbf{P}^*$ is the optimal transport matrix.

the TCRdist values between them, with a few distances shown in green for $t_{2,3}$ (the rest are omitted for brevity).

## Effort and loneliness

Let $\widehat{\mathbf{P}}$ be an estimate of the optimal transport matrix (such as the Sinkhorn approximation $\mathbf{P}^\lambda$) between repertoires $R_1$ and $R_2$, with corresponding distance matrix $\mathbf{D}$. Define the "effort" matrix as the Hadamard product of $\widehat{\mathbf{P}}$ and $\mathbf{D}$,

$$\mathbf{E} \coloneqq \widehat{\mathbf{P}} \odot \mathbf{D} = (\widehat{\mathbf{P}}_{ij}\mathbf{D}_{ij})_{1 \leq i \leq n, 1 \leq j \leq m} \in \mathbb{R}^{n \times m}. \tag{7}$$

For any $t_i \in R_1$, $t_j \in R_2$, define

$$\text{PairedEffort}(t_i, t_j) \coloneqq \mathbf{E}_{ij} \equiv \widehat{\mathbf{P}}_{ij}\,\text{TCRdist}(t_i, t_j) \tag{8}$$

which can be interpreted as the entrywise amount of "effort" or "work" used in the optimal transport matrix to move the mass at TCR $t_i$ to TCR $t_j$.

We wish now to define a score that quantifies the isolation of a given TCR in one repertoire relative to some reference repertoire, where a high score indicates that the TCR is characteristic of its own repertoire but unusual with respect to the reference repertoire. A naive score for a given TCR $t_2 \in R_2$ with respect to all the TCRs in $R_1$ would be

$$\text{IndividualLoneliness}(t_2 \mid R_1) = \sum_{t_1 \in R_1} \text{PairedEffort}(t_1, t_2) \tag{9}$$

which reduces to a sum of the column of $\mathbf{E}$ that indexes $t$. A drawback of Eq (9) is that there might be outlier TCRs in $R_2$ that also look lonely to $R_1$ as a result, and thus would yield a high loneliness value. We are interested in a more differential version of loneliness: a TCR that is lonely with respect to a different repertoire $R_1$ but not very lonely with respect to its own repertoire $R_2$ (as illustrated in Fig 1). This would suggest that $t$ is indicative of some feature of $R_2$ not present in $R_1$ (e.g. a vaccination).

Instead, we consider the cumulative individual loneliness around a neighborhood of size $\delta$ around each $t$:

$$\text{NeighborhoodLoneliness}(t_2 \mid R_1; \delta) := \sum_{t' \in \mathcal{B}_\delta(t_2)} \text{IndividualLoneliness}(t' \mid R_1) \tag{10}$$

where $\mathcal{B}_\delta(t_2) = \{t' : \text{TCRdist}(t_2, t') < \delta\}$. This reduces to a sum of all columns indexing some TCR in $\mathcal{B}_\delta(t_2)$. The *neighborhood loneliness* Eq (10) will be small when $t$ is an outlier in both repertoires, since there won't be many neighboring TCRs $t'$ in the ball. Further, Eq (10) will be large for a TCR with many neighbors in $R_2$ but few in $R_1$, since there will be many nearby TCRs all with relatively high transport values. Because of these properties, we use Eq (10) as the core scoring mechanism for our methods and analyses presented here, and will simply call this value *loneliness*. The expression in Eq (10) relies on a neighborhood radius parameter $\delta$ which requires tuning: setting $\delta$ too small will lead to unstable results as there will rarely be neighbors for a given $t$, and setting $\delta$ too large will assign too many neighbors to each TCR and grossly inflate the scores. However, we show that Eq (10) consistently behaves better than Eq (9) as an indicator of lonely groups of TCRs for each sensible radius $\delta$. For example, Eq (10) does a significantly better job discriminating "spiked-in" epitope-specific TCRs from a background of naive TCRs in simulation experiments (see S1 Fig and the results section on simulation-based benchmarking).

## Clustering

We wish to identify collections of similar TCRs that appear to be enriched in their own repertoire relative to a reference repertoire using our loneliness scores defined in Eq (10). Suppose we have computed loneliness values for TCRs $t_1, \ldots, t_m \in R_2$ with respect to a source repertoire $R_1$. We first describe a procedure to identify the "loneliest" cluster in $R_2$, and then show how iterating this scheme will allow us to compute any remaining lonely clusters.

Start with the loneliest TCR $t_{\max}$ and some step size $s$ (by default, we choose $s = 5$). In each iteration $i$, step out $s$ units of TCRdist from previous radius $r_{i-1}$, and compute the mean loneliness of all TCRs within $r_{i-1}$ and $r_{i-1} + s$ units of $t_{\max}$:

$$S_i := \{\tau : r_{i-1} \leq \text{TCRdist}(t_{\max}, \tau) < r_{i-1} + s\} \tag{11}$$

$$m_i := \frac{1}{|S_i|} \sum_{t \in S_i} \text{NeighborhoodLoneliness}(t \mid R_1). \tag{12}$$

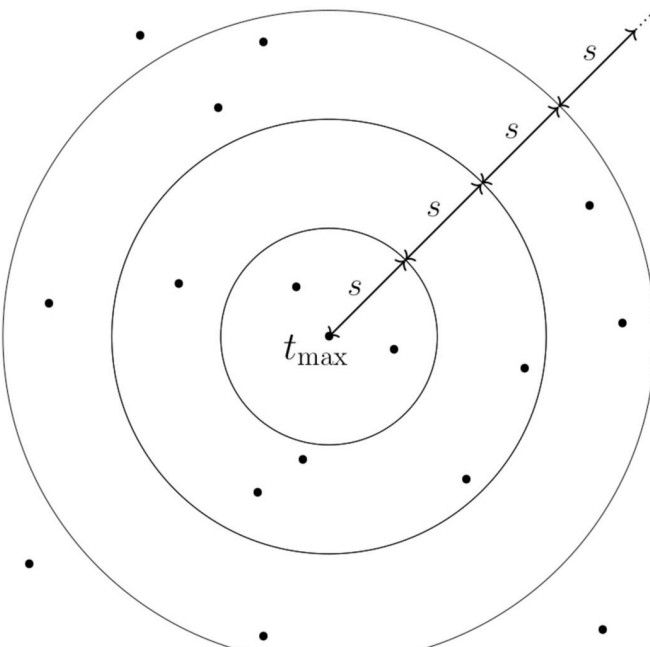

**Fig 3. A schematic of our clustering procedure in Algorithm 1.** Each point is a TCR portrayed in an abstract 2-D space, where the distance between points is determined by TCRdist. Our procedure starts by identifying the maximally lonely TCR $t_{\max}$ according to Eq (10). In each iteration, we step out $s$ units of TCRdist, and compute the mean loneliness of all TCRs within the annulus defined by the current and previous radii (or ball in the first step). By construction of Eq (10), we expect the loneliness values to steadily decrease as we move away from $t_{\max}$, until we arrive at a radius where the loneliness values have stabilized. This "breakpoint radius" thus defines the radius of our cluster.

In the first iteration, we are just looking at the mean loneliness in the TCRdist ball of $s$ units around $t_{\max}$. After that, each iteration looks at the mean loneliness of the semi-closed annulus of width $s$ surrounding the previous region. Once we have have computed values of Eq (12) for our full set of radii (e.g., $r = 0, 5, 10, \ldots, 200$), we can examine the relationship of mean loneliness vs radius to see if there is a breakpoint $r_{\text{breakpoint}}$ at which loneliness is no longer high. If a breakpoint is detected, we simply define our cluster as those TCRs which fall within $r_{\text{breakpoint}}$ units of TCRdist to $t_{\max}$. This procedure is illustrated in Fig 3.

The above procedure yields a cluster of the "loneliest" TCRs of our full repertoire. To identify further lonely clusters, we simply identify the loneliest TCR that has not yet been clustered, and apply the same procedure. We can iterate until a sensible stopping point, such as when a breakpoint is unable to be estimated (discussed further in the next paragraph). The complete algorithm for a maximum number of clusters $C$ is formalized in Algorithm 1. To gain intuition about how the algorithm works, and as a visual confirmation of the assumptions behind the algorithm, we refer the reader to S2 Fig.

To estimate breakpoints in the above procedure, we perform a segmented regression, also known as a piecewise regression, of the mean annulus loneliness values $m_i$ on the set of radii $r_i$. Univariate segmented regression assumes that the relationship between the response and predictor is described by a pair of differing line segments across two separate intervals separated by a breakpoint $\rho$, with the line segments coinciding at $\rho$. For our response $M$, the mean annulus loneliness, with a fixed cluster radius $r$ as our predictor, our

model becomes

$$\mathsf{E}[M \mid r] = \beta_0 + \beta_1 r + \beta_2 (r - \rho)_+ \tag{13}$$

$$= \begin{cases} \beta_0 + \beta_1 r, & r \leq \rho \\ (\beta_0 - \beta_2 \rho) + (\beta_1 + \beta_2) r, & r > \rho \end{cases}. \tag{14}$$

Here, $x_+ = x$ if $x > 0$, and 0 otherwise. The least squares method yields estimates $\widehat{\beta}_0, \widehat{\beta}_1, \widehat{\beta}_2$, and $\widehat{\rho}$ of the corresponding model parameters $\beta_0, \beta_1, \beta_2$, and $\rho$. We can then use $\widehat{\rho}$ as our estimate of the breakpoint radius $r_{\mathrm{breakpoint}}$, and define our cluster as $\{t : \mathrm{TCRdist}(t, t_{\mathrm{max}}) \leq \widehat{\rho}\}$. We use the segmented R package to estimate the coefficients of these models for our analyses [35]. Note that a breakpoint may be unable to be estimated if the regression assumptions are not met well (there is no strong evidence of an "elbow" from the data), the initial breakpoint is not close enough to the "true" breakpoint, or there are not enough data to estimate the model parameters. Nonetheless, we verify the desired behavior of this regression approach to estimate cluster radii using a set of typical TRB datasets in Appendix A.

**Algorithm 1** Computing clusters of lonely TCRs

**Input:** Repertoires $R_1$ and $R_2$, radius step size $s > 0$, maximum cluster count $C \geq 1$
**Output:** Vector of TCR clusters $\mathbf{c} = [c_1, \ldots, c_C]$ of $R_2$
1: $\mathbf{c} = []$
2: keep_clustering $\leftarrow$ True
3: **while** keep_clustering **do**
4:   $R_2^{\mathrm{sub}} \leftarrow \{t_2 \in R_2 : t_2 \text{ is not clustered}\}$ ▷ get all un-clustered TCRs
5:   $t_{\mathrm{max}} \leftarrow \max_{t_2 \in R_2^{\mathrm{sub}}} \mathrm{NeighborhoodLoneliness}(t_2; R_1)$ ▷ get loneliest un-clustered TCR
6:   $r_{\mathrm{prev}} \leftarrow 0$
7:   $r_{\mathrm{current}} \leftarrow s$
8:   **while** $r \leq r_{\mathrm{max}}$ **do**
9:     $S \leftarrow \{t: r_{\mathrm{prev}} < \mathrm{TCRdist}(t, t_{\mathrm{max}}) \leq r_{\mathrm{current}}\}$ ▷ define annulus
10:     $\ell_r \leftarrow \mathrm{mean}_{t \in S} \mathrm{NeighborhoodLoneliness}(t; R_1)$ ▷ compute mean loneliness over all TCRs in annulus
11:     $r_{\mathrm{prev}} \leftarrow r_{\mathrm{current}}$
12:     $r_{\mathrm{current}} \leftarrow r_{\mathrm{current}} + s$ ▷ update annulus radii
13:   **end while**
14:   estimate breakpoint $r_{\mathrm{breakpoint}}$ of $\ell_r$ vs. $r$
15:   **if** $r_{\mathrm{breakpoint}} = \mathrm{NULL}$ **then** ▷ we were unable to estimate a cluster radius, so terminate
16:     keep_clustering $\leftarrow$ False
17:   **else** ▷ we succeeded in detecting a cluster radius
18:     append $\{t: \mathrm{TCRdist}(t, t_{\mathrm{max}}) \leq r_{\mathrm{breakpoint}}\}$ to $\mathbf{c}$ ▷ append our cluster to the running vector
19:     **if** $|\mathbf{c}| = C$ **then** ▷ we have detected the maximum # of clusters
20:       keep_clustering $\leftarrow$ False
21:     **end if**
22:   **end if**
23: **end while**
24: **return c**

## Motif inference

Given a cluster of TCRs, we would like to infer a motif describing their sequence homology, however CDR3 lengths can vary by TCR. One solution is to generate a regular expression that describes the sequences in the cluster, either manually or through some algorithm. However, this can be messy/difficult, and raw regular expressions are not always easily interpretable by eye.

Instead, we appeal to profile-HMMs, which describe the emission probabilities of amino acids at each position along a sequence while explicilty modeling position-specific insertion and deletion probabilities [36]. Profile-HMMs are easily used with the HMMer package (http://hmmer.org/). Estimating a profile-HMM $\pi$ allows for several benefits.

- We can model an arbitrary cluster of TCR sequences with $\pi$ without worrying about CDR3 length differences

- We can query other sequences against this profile to assess their homology to the cluster in a statistically rigorous manner. In particular, for any animo acid sequence $\sigma$, we can first compute a log-odds ratio "bit score" comparing the likelihood of observing $\sigma$ from $\pi$ to the "null" likelihood of observing $\sigma$ from an independent, identically distributed random sequence model $\pi_0$. Then, we can compute an E-value which is based on the number of hits expected to achieve this bit score or greater by chance, i.e. if the search had instead been done using $\pi_0$. We further use these E-values to define "hard" motif memberships via the indicator $\mathbb{1}(e < e_{\mathrm{crit}})$, for some specified critical threshold $e_{\mathrm{crit}}$.

- We can readily visualize these profiles via enhanced sequence logos that display indel characteristics [37, 38]

As HMMer requires aligned sequences in order to build HMMs, we use MAFFT, a fast and popular tool that constructs a multiple-sequence alignment of a given set of query sequences [39], whenever we need such alignments for HMMer.

## Significance estimates

We wish to attach significance estimates to our loneliness scores to determine whether high observed scores are improbable due to chance alone. For this, we perform the following randomization test given repertoires $R_1$ and $R_2$. For each trial $j \in \{1, \ldots, J\}$, randomly re-label the TCRs in $R_1$ and $R_2$ to get trial repertoires $\tilde{R}_1^{(j)}$ and $\tilde{R}_2^{(j)}$ (each TCR is independently reassigned, even if it appears multiple times). Under the null hypothesis that $R_1$ and $R_2$ are samples from the same (abstract) population repertoire of TCRs, each of these trial repertoires $\tilde{R}_1^{(j)}$ and $\tilde{R}_2^{(j)}$ will have the same sampling distribution as $R_1$ and $R_2$. We then compute Eq (10) for each TCR in $\tilde{R}_2$ with respect to $\tilde{R}_1$. After $J$ trials, we have obtained null distributions of loneliness scores for each $t \in R_2$. We can now compare the observed loneliness $\ell_{\mathrm{obs}}$ of a given TCR $t$ to its null distribution, rejecting the null hypothesis if $\ell_{\mathrm{obs}}$ is sufficiently high (e.g., higher than the $1 - \alpha$ quantile for a specified level $\alpha$).

There are a couple of caveats to this approach. First, when we relabel TCRs during a given trial $j$, only some of the TCRs from $R_2$ will be present in $\tilde{R}_2^{(j)}$. We handle this by maintaining score distributions only for the TCRs originally present in $R_2$, and appending trial scores for only those TCRs to their running distributions (thus ignoring the scores of TCRs in $\tilde{R}_2^{(j)}$ that originally belonged to $R_1$). After $J$ trials, we downsample these score distributions to the size of the smallest distribution. If there is a particular minimal sample size we desire for all of the

score distributions, we could simply add a check in our routine to stop only when this minimal sample size has been attained (although this would lead to an increased runtime).

## Data

The following TCR$\beta$ repertoire datasets are used in the above analyses.

The majority of our analyses involve TCR$\beta$ repertoires collected from 23 C57BL/6 mice [40], which form biological replicates for our study. For each mouse subject, three repertoires were sampled, corresponding to their CD4$^+$, CD8$^+$, and double negative (DN) intraepithelial lymphocyte (IEL) repertoires. In terms of receptors, the group of cells we call DN are CD4$^-$ CD8$\alpha\beta^-$ CD8$\alpha\alpha^+$, which are distinct from the class of preselection T cells which are sometimes also called "DN".

Thus, there are $23 \times 3 = 69$ total IEL mouse datasets, to which we collectively refer as the *IEL data*. We will typically abbreviate CD4$^+$ as "CD4", and CD8$^+$ as "CD8". For brevity, we will define the collection of datasets for a given IEL type as a subscripted $\mathcal{R}$. For example, $\mathcal{R}_{CD4}$ denotes the collection of 23 CD4$^+$ repertoires. Each repertoire consists of TCR sequences described by a V gene and CDR3aa pair (i.e., J genes are excluded from the analysis). The sequence data preprocessing for this study was performed using MIGEC [41]; more details can be found in [40].

Our second analysis examines TCR$\beta$ repertoires collected from six human donors before and after an immunization with live yellow fever virus (YFV) vaccine [20]. Samples were taken from each donor at multiple timepoints: 7 days prior to vaccination (−7d), the day of vaccination (0d), 15 days following vaccination (+15d), and 45 days (+45d) following vaccination. This yields $6 \times 4 = 24$ human YFV datasets, to which we will collectively refer as the *YFV data*. The processed TCR repertoires from Ref. [20] were downloaded from https://github.com/ mptouzel/pogorelyy_et_al_2018 and filtered to the 1,000 most abundant clones. As for the IEL data, each repertoire consists of TCR sequences described by a V gene and CDR3aa pair.

Our final analysis examines the ability of the optimal transport framework to discriminate "spiked-in" epitope-specific TCR sequences from a larger repertoire of naive CD8+ TCR sequences, by comparing the mixed repertoire to a background repertoire consisting only of naive CD8+ TCR sequences. The naive CD8+ TCR sequences were randomly selected from a public dataset released by 10X Genomics [42]. The "spike-in" TCR sequences are specific for the influenza M1$_{58}$ epitope presented by HLA-A$^*$02:01 and were collected from the literature and public databases by Schattgen et al [43].

## Implementation

Our Python implementation of TCR optimal transport analysis, using the Python Optimal Transport [44] package, can be found at https://github.com/matsengrp/transport. It uses the C++ implementation https://github.com/phbradley/pubtcrs/ of TCRdist. The repository includes straightforward instructions on how to reproduce the analyses in this paper by running a script.

For the analyses in this paper, we used a neighbor radius of 48.5 and a Sinkhorn regularization of 0.01. A TCRdist of 48 might correspond to some combination of 0–4 mismatches in the CDR3 loop together with 0–12 mismatches in the other CDR loops (depending on the chemical nature of the amino acid differences). This choice of neighbor radius appears to work well in practice for single-chain data, rewarding neighborhood density while retaining specificity of sequence features in local neighborhoods. For paired data, a larger radius value of 100–120 would be more appropriate.

By default, each independent input TCR sequence is given equal probability mass in the transport analysis. For input files that have been reduced to unique nucleotide-level clonotypes, as in the present manuscript, this has the effect of ignoring clonal abundance (except when subsetting to the top expanded clonotypes, as in the YFV analysis). The extent of clonal abundance (i.e., the numerical sizes of expanded clonotypes) can be noisy and sequencing-method dependent, which makes this a more conservative approach: significant differences in TCR landscape density are driven by accumulation of independent rearrangement/selection events rather than individual clonal expansions. For situations in which one wants to use clonal abundance, it is straightforward to re-duplicate clonotypes prior to input to the pipeline, which will assign additional mass proportional to the number of copies of each clonotype.

We found that a majority of the runtime for our approach is devoted to the TCRdist calculation, even using the C++ implementation (S3 Fig). To establish this, we measured the pairwise run times of TCRdist and OT calculation on each of the IEL DN (background) replicates against every other background replicate, as well as against all CD4 (foreground) replicates. Across the total 1058 paired calculations, we found that TCRdist run times took an average of 2.42e-01 seconds, while optimal transport calculation had a lower average run time of 1.05e-01. These measurements were performed on a node running Ubuntu 18.04 (bionic) with 16 Intel Xeon CPU E5–2667 v4 @ 3.20GHz, and 256GB RAM.

## Results

We have defined a "loneliness" measure which captures TCRs that are characteristic of their own repertoire but unusual with respect to a reference repertoire. We have also presented a procedure to obtain clusters of sequences without equivalents in the reference repertoire based on these loneliness scores. In this section we apply our loneliness and clustering methodology to the IEL data and the YFV data, and show that our clusters capture meaningful differences between repertoires that we know are sampled from distinct populations.

### Consistent loneliness dynamics across biological replicates of IEL mice

In this section we examine the behavior of our loneliness scores defined by Eq (10) in the context of the IEL data described above. The IEL data contain TCR repertoires of three distinct cell types, referred to as CD4, CD8 and DN ("double negative" CD4$^-$ CD8$\alpha\beta^-$ CD8$\alpha\alpha^+$) cells, from 23 genetically identical mice. These cell types differ in their expression of certain receptor proteins and their interactions with other cells, which impacts the binding properties of their corresponding TCR repertoires. Thus, we expect there to be meaningful differences in their respective TCR sequence distributions. In our analysis, we will focus on identifying TCRs in the DN repertoire that are characteristic of the DN repertoire, but are unusual with respect to the CD4 repertoire. This will allow us to use the CD8 repertoire as a useful comparison set since it will not influence the loneliness scores. Nonetheless, analogous analyses could be performed between any two cell types, with the third cell type available for comparison.

This data set has 23 sampled biological replicates for each cell type, which allows us to understand the true biological variability of observing a given TCR in a sample. This provides us with a robust representation of each of the CD4, CD8, and DN population repertoires for our comparisons. In particular, we can get a sense of the overall differences between the DN and CD4 TCR$\beta$ sequences by combining each of the respective sets of repertoires into two large, representative datasets. Specifically, we concatenate $R_{DN-1}, \ldots, R_{DN-23}$ to obtain a combined DN repertoire $R_{combined-DN}$, and we concatenate $R_{CD4-1}, \ldots, R_{CD4-23}$ to obtain a combined CD4 repertoire $R_{combined-CD4}$.

Next, we compute NeighborhoodLoneliness($t$; $R_{\text{combined-CD4}}$) for each $t \in R_{\text{combined-DN}}$ to score each DN TCR $t$ given the landscape of CD4 TCRs represented by $R_{\text{combined-CD4}}$. We also apply Algorithm 1 to $R_{\text{combined-DN}}$ with these loneliness scores to compute the top several lonely clusters. These clusters are constructed to be centered around the most lonely TCRs, and to encompass the surrounding similarly lonely TCRs within an estimated TCRdist cutoff. We expect these top clusters to represent significant differences between the CD4 and DN repertoires, since they should by construction contain the loneliest TCRs that reside in sufficiently dense regions of the TCR landscape obtained from the combined DN repertoires. We will refer to these top three loneliest clusters as the OT-Tremont, OT-Revere, and OT-Ida clusters, respectively. The OT-Tremont and OT-Revere clusters are so named because of their high similarities to the Tremont and Revere clusters described in Figure 5M of [40]. While the authors of [40] present detailed motif specifications, the Tremont cluster is dominantly characterized by the GT[VI]SNERLFF CDR3$\beta$aa motif, and the Revere cluster consists of a TRBV16 gene paired with a dominant DWG CDR3$\beta$aa motif. The OT-Ida cluster represents a novel TCR motif specification to the best of the authors' knowledge.

V gene usage and CDR3aa motifs for our three clusters are visualized in Fig 4. Each CDR3aa motif is visualized with a profile-HMM sequence logo obtained from Skylign [38]. The height of each stack is proportional to the level of conservation at that position, and the height of each amino acid within a stack is proportional to the probability of observing that amino acid at that position. The first row of numbers below the sequence logo displays each position's occupancy, or the probability of observing a non-gap character at that position (so that (1−occupancy) gives the position's deletion probability). The second row displays the insertion probabilities at the respective positions, so that the $k$th value represents the probability of an insertion between positions $k$ and $k + 1$. The third row displays the expected insertion lengths of an insertion following position $k$, if an insertion exists.

Each cluster has distinctive features which suggest conservation of particular V genes and/ or CDR3 amino acid motifs. The OT-Tremont cluster has the strictest V gene profile, containing only the TRBV16*01 gene. It also seems to include a conserved subsequence roughly spanning positions 5–8 in the sequence logo, as well as positions 11–17 which likely correspond to a stringent J gene specification. The OT-Revere cluster has a relatively loose V gene profile, containing 18 TRBV genes total, though the TRBV12–1*01 and TRBV12–2*01 genes comprise the majority of TRBV genes in this cluster. However, this cluster seems to have notable levels of conservation across most or all of the CDR3 sequence, with position 8 being the only one without a clearly dominant amino acid. The OT-Ida cluster has a strict V gene profile, with over 90% of the sequences consisting of the TRBV12–1*01 or TRBV12–2*01 genes. There is also evidence of varying levels of conservation throughout the CDR3, with only a few positions (6, 9, 10) lacking a dominant amino acid.

These automatically-generated clusters are able to capture subsets of the DN repertoires that are distinguished from the CD4 repertoires (Fig 5). To see how often each cluster is observed among the different individual DN, CD4, and CD8 repertoires, we can plot frequency polygons of each cluster prevalence empirical distribution (Fig 5A), where the empirical prevalence of a cluster $m$ within a repertoire $R$ is defined as

$$\widehat{\text{Prevalence}}(c) := \widehat{\text{Pr}}_{T \sim R}(T \in c) \tag{15}$$

$$= \frac{1}{|R|} \sum_{t \in R} \mathbb{1}(t \in c). \tag{16}$$

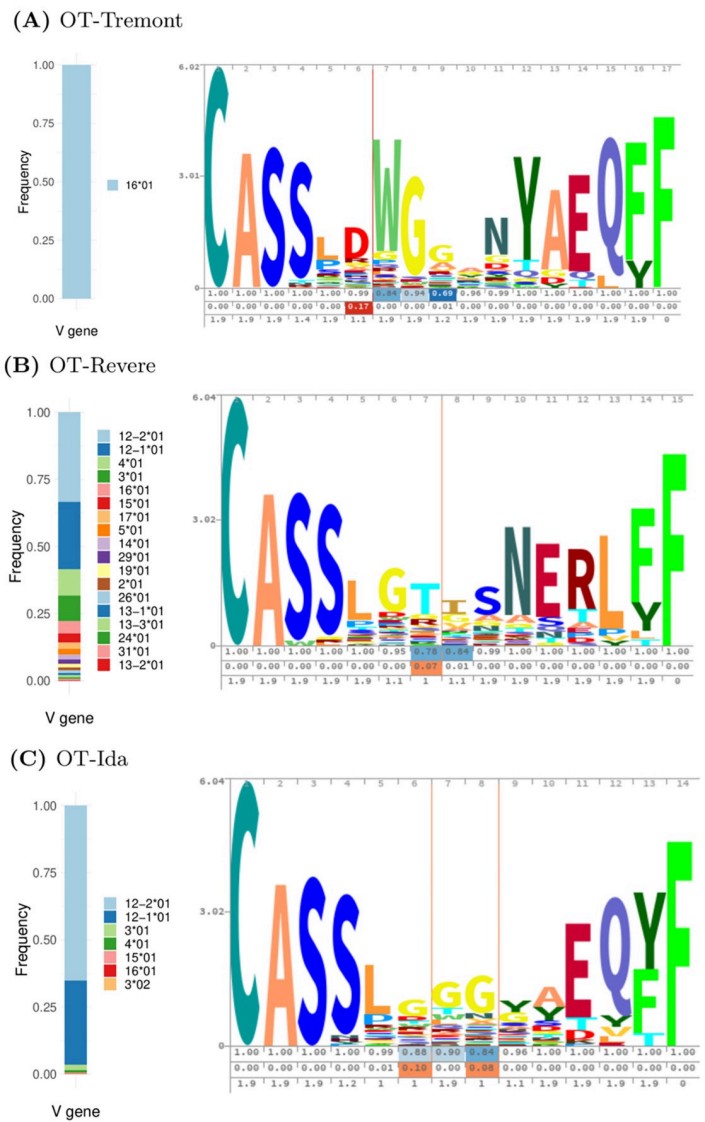

**Fig 4. Visualizations of TRBV gene frequency statistics and CDR3aa sequence logos for the top three lonely clusters of the combined repertoire analysis: (A) OT-Tremont, (B) OT-Revere, (C) OT-Ida.** The height of each stack within the sequence logo is proportional to the level of that position's conservation, and the height of each amino acid is proportional to that amino acid's frequency in that position. The rows below each sequence logo display the occupancies, insertion probabilities, and expected insertion lengths of the respective positions.

We see that each cluster tends to have higher prevalences in DN distributions compared to CD4 distributions, which indicates that these clusters are enriched in the DN population with respect to the CD4 population. For the OT-Tremont cluster, the prevalence is almost always zero for CD4 repertoires yet nonzero for each DN repertoire. For the OT-Revere and OT-Ida clusters, there are more nonzero prevalences in the CD4 repertoires, but consistently higher prevalences in the DN repertoires. Interestingly, the OT-Tremont and OT-Revere clusters tend to have similar prevalences among the CD4 and CD8 repertoires, whereas the OT-Ida cluster tends to have similar prevalences among the CD8 and DN repertoires. This matches the intuition behind our scores defined by Eq (10), which seeks to identify enrichment of

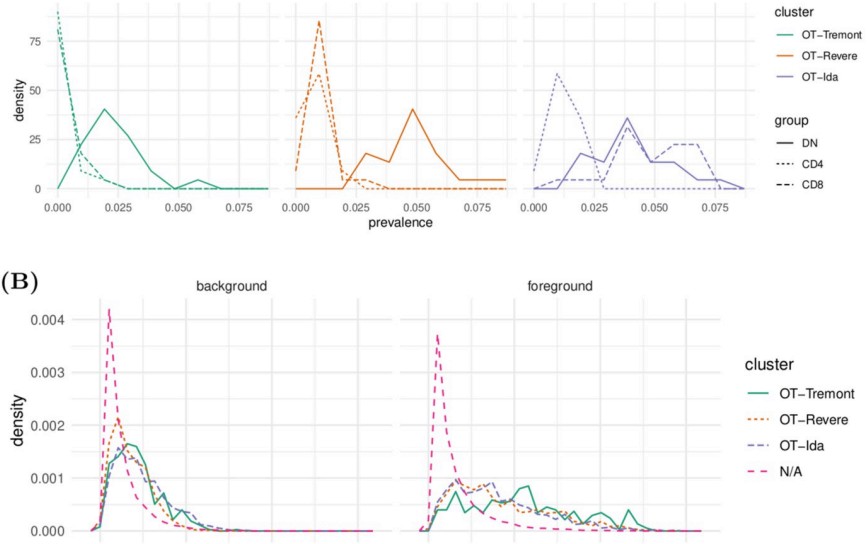

**Fig 5. Plots of several statistics that describe the across-repertoire dynamics of the OT-Tremont, OT-Revere, and OT-Ida clusters.** (A) Distributions of cluster prevalence across combined repertoires, stratified by cluster and cell type. The DN distributions are consistently to the right of the CD4 distributions, showing that our algorithm is finding motifs that are highly represented in the DN repertoire compared to the CD4 repertoire. (B) Distributions of neighborhood loneliness scores across individual repertoires, stratified by cell type group (background/foreground) and cluster. N/A means not assigned to a cluster. We again see that the motifs distinguish DN sequences from CD4 sequences above the level of per-repertoire variation.

subsets of DN repertoires with respect to CD4 repertoires, and not with respect to any arbitrary null distribution.

To get a sense of the loneliness dynamics of these clusters that does not rely on the scores obtained from the combined repertoires above, we performed the following experiment. Recall that each of these combined repertoires consists of a collection of independent repertoire samples. For a given DN repertoire, we define the background set as all other DN repertoires, and the foreground set as all CD4 repertoires. The idea is that there will be intrinsic variability, or "background noise", that can be observed between repertoires of a common cell type, whereas two different cell types will also possess "foreground" variability that corresponds to biological differences between the repertoires. For each DN repertoire $R_{\text{DN}}$, and each $t \in R_{\text{DN}}$, compute the following background and foreground scores:

$$\text{bg-score}(t) := \frac{1}{|\mathcal{R}_{\text{DN}}| - 1} \sum_{R \in \mathcal{R}_{\text{DN}} \setminus R_{\text{DN}}} \text{NeighborhoodLoneliness}(t \mid R) \tag{17}$$

$$\text{fg-score}(t) := \frac{1}{|\mathcal{R}_{\text{CD4}}|} \sum_{R \in \mathcal{R}_{\text{CD4}}} \text{NeighborhoodLoneliness}(t \mid R) \tag{18}$$

We can interpret Eq (17) as the average neighborhood loneliness of a given DN TCR with respect to the background set of all other DN repertoires, and Eq (18) analogously but with respect to the foreground set of all CD4 repertoires. We expect these averages to give stable estimates of how lonely this TCR looks compared to either of these two populations.

Distributions of scores obtained from Eqs (17) and (18), stratified by cluster, are shown in Fig 5B, in the left and right panels respectively. We see that these scores of each specified cluster tend to be higher than TCRs without a specified cluster for the background set, and these scores become amplified in the foreground set. This indicates that TCRs belonging to these clusters consistently have higher loneliness values compared to CD4 repertoires versus DN repertoires, as expected.

The results above demonstrate that the three clusters identified by our algorithm applied to the combined repertoires $\mathcal{R}_{DN}$ and $\mathcal{R}_{CD4}$ have amplified prevalences in the individual DN repertoires with respect to the CD4 repertoires, and yield consistently high loneliness scores across the individual replicate repertoires. This indicates that our algorithm is able to detect clusters of TCRs which seem to be differentially enriched between the subpopulations in question.

The total optimal transport between two TCR repertoires represents an intuitive and theoretically well-founded metric of global repertoire dissimilarity that can be used for applications such as repertoire clustering. As an illustration, we calculated the matrix of total optimal transport distances between the TCR$\beta$ repertoires of the IEL dataset and performed hierarchical clustering with Ward's linkage criterion. Examination of the clustering dendrogram (Fig 6) revealed that the repertoires cluster according to cell subpopulation, with two exceptions: the CD8 repertoire from mouse 19, one of the smallest repertoires with 80 sequences, was grouped with the CD4 repertoires, and the CD8 repertoire from mouse number 22 was grouped with

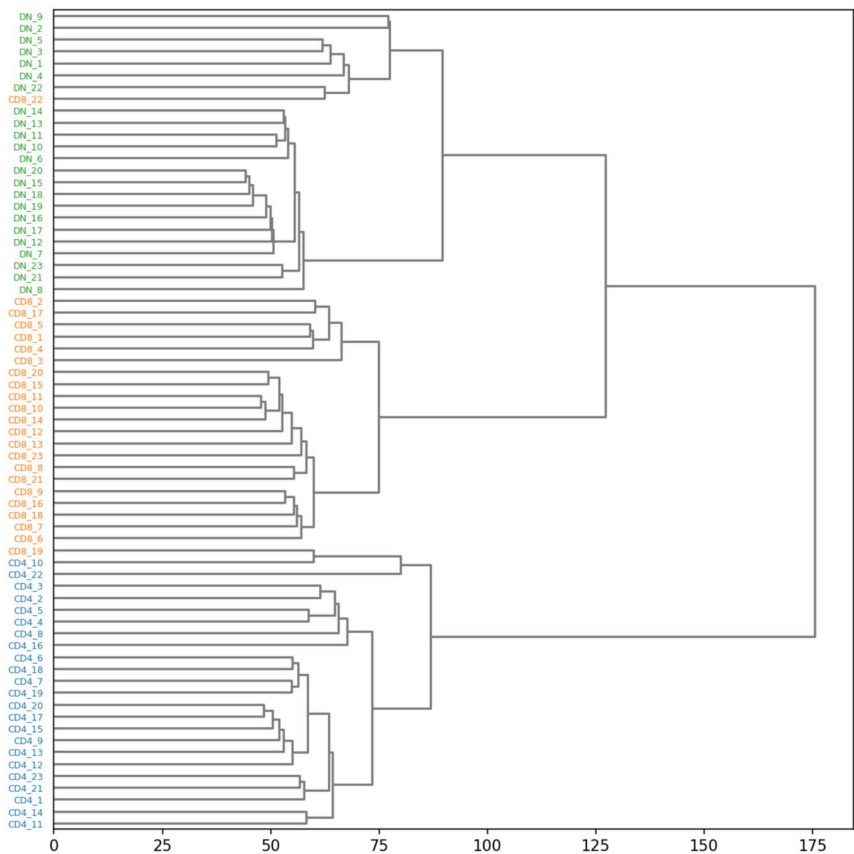

**Fig 6. A hierarchical clustering tree built from the matrix of optimal transport repertoire distances, with the tips colored by the cell subpopulation: Green for DN, orange for CD8, and blue for CD4.**

the DN repertoire from the same mouse. These latter two repertoires shared a higher fraction of sequences than any other pair of repertoires, a possible indication of minor contamination during cell sorting. Overall, the optimal transport repertoire clustering analysis accords well with biological intuition while also providing potentially valuable information on similarity relationships between specific repertoires.

## Validating randomization test scores with biological replicates

The randomization test framework presented earlier aims to determine how statistically significant an observed loneliness score is compared to what we would expect under a null model of having no significant differences between repertoires. The efficacy of this test depends on how accurately the randomization distributions mimic the dynamics of two repertoires that are truly sampled from the same population. We can benchmark this using the IEL biological replicates, since these replicates are samples from a common population of genetically identical mice; this allows us to quantify the statistical characteristics of the resultant TCR distributions, and consequently, the loneliness score distributions. If the statistical characteristics of these true replicate loneliness distributions approximately match the statistical characteristics of the "randomization" loneliness distributions, this gives us confidence in our testing procedure and significance estimates as they perform on real data.

We apply the randomization test framework described in the methods section below to the IEL replicates as follows. First, we identify the largest DN TCR$\beta$ repertoire $R_{DN}$ (subject #15; 1,737 sequences) and largest CD4 TCR$\beta$ repertoire $R_{CD4}$ (dataset # 17; 864 sequences). We chose the largest dataset in hopes of obtaining the most stable parameter estimates, although relative and absolute sample sizes did not seem to majorly contribute to the behavior across various combinations of repertoires. For each $t \in R_{DN}$, we compute the observed score $s_{obs} =$ NeighborhoodLoneliness($t; R_{CD4}$) using Eq (10). Then, we compute the distribution of score values for $t$ across the "background distribution" of all other DN repertoires,

$$S(t) := \{\text{NeighborhoodLoneliness}(t; R') : R' \in \mathcal{R}_{DN} \setminus R_{DN}\}. \tag{19}$$

Since the set of DN repertoires are biological replicates, we can use $S(t)$ as a null distribution of $s_{obs}$. In particular, we can compute a "replicate" $z$-score

$$z(t) = \frac{s_{obs} - \text{mean}(S(t))}{\text{stddev}(S(t))} \tag{20}$$

to quantify how surprising the observed $s_{obs}$ is with respect to the replicate null distribution (this also allows for the computation of $p$-values).

Next, we apply our randomization procedure to the same ($R_{DN}, R_{CD4}$) pair, to get a randomization distribution of score values

$$S^\star(t) := \{\text{NeighborhoodLoneliness}(t; R^\star)\} \tag{21}$$

where $R^\star$ ranges over our set of randomization datasets mixing the pair ($R_{DN}, R_{CD4}$).

We can compute a corresponding "randomization" $z$-score

$$z^\star(t) = \frac{s_{obs} - \text{mean}(S^\star(t))}{\text{stddev}(S^\star(t))}. \tag{22}$$

If our randomization procedure produces a reliable approximate null distribution of $z$-scores, and if our set of biological replicates approximate the sampling distribution of DN TCR$\beta$ sequences well, then we would expect there to be a clear relationship between $z(t)$ and $z^\star(t)$.

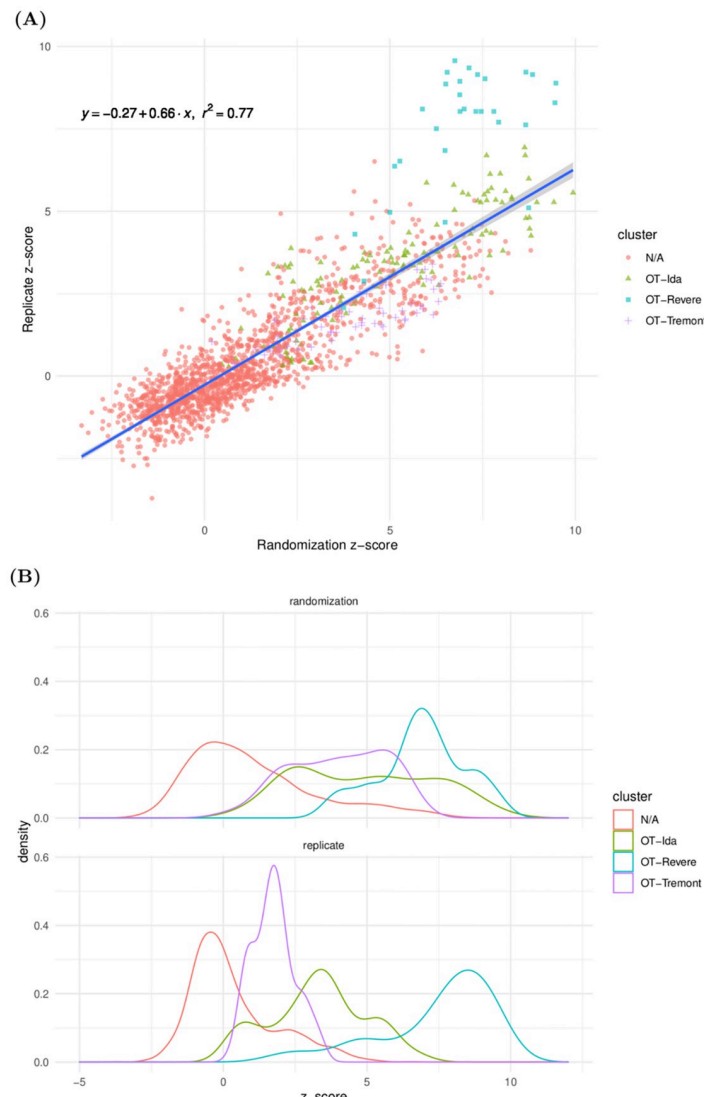

**Fig 7. Visualizations of the relationship between replicate and randomization z-scores.** (A) Scatterplot of replicate z-scores versus randomization z-scores. (B) Marginal density estimates of replicate z-scores and randomization z-scores. We see strong evidence of a significantly positive linear relationship between these quantities, suggesting that our randomization procedure is able to identify significant differences between repertoire datasets.

We see that the randomization $z$-scores $z^\star$ and replicate $z$-scores $z$ exhibit a strong linear relationship (Fig 7A), with a correlation coefficient of $\rho \approx 0.877$. Further, we can assess how our randomization null distribution behaves as a proxy to the biological replicate null distribution by performing a standard linear regression of $z$ on $z^\star$. Because the scatterplot reveals clear heteroskedasticity, we use sandwich estimation to obtain robust standard error estimates. This yields a significantly positive slope coefficient of $\beta \approx 0.656259$ ($p < 2 \times 10^{-16}$), with relatively high predictiveness (adjusted $R^2$: 0.769, $p < 2.2 \times 10^{-16}$). We note that there is some visual evidence that the relationship might exhibit nonlinearity in the right tail, particularly due to the OT-Revere cluster which seems to mostly reside above the regression line. Nonetheless, we believe this model is still useful to understand the strength and general behavior of the relationship.

We emphasize that the slope of this line is not one, and so the randomization p-value can not be interpreted directly as a significance test where the null is a biological replicate in a different organism. This is not surprising, since there is genuine biological variation between the various mice. If one desired to approximate a between-organism p-value, one could use the values of the linear regression to map the randomization Z-score to a replicate Z-score and calculate a p-value accordingly.

We also examine marginal density estimates of these z-scores stratified by cluster (Fig 7). For the three named clusters, the densities appear to have means notably higher than zero.

In summary, the approximate null distribution of loneliness scores from our randomization test appears to accurately represent the loneliness score distribution under the null hypothesis of two repertoires representing the same underlying population. Furthermore, we see that both the randomization null distribution and the replicate null distribution both lead to significantly high loneliness scores for the top three lonely clusters identified above. This indicates that we can confidently obtain significance estimates for loneliness scores when comparing two TCR repertoires.

## Identifying responsive TCRs to a yellow fever vaccination

Next, we benchmark the ability of our methods to detect meaningful differences between longitudinal repertoires using the YFV data discussed in the Materials and methods section. In particular, for each of the six human donors, we perform three comparisons: −7d vs 0d, 0d vs +15d, and 0d vs +45d. For each comparison, we compute the top 10 loneliest clusters using Algorithm 1. Since the immune response was estimated to peak at day 15 for all subjects and had contracted by day 45, we expect there to be many responsive TCRs in the +15d repertoire vs 0d, and we expect the number of false positives to increase for lower-ranked (i.e., less lonely) clusters. We expect some residual responsive TCRs in the +45d repertoire but with lower levels than the +15d repertoire. Finally, we expect little to no responsive TCRs in the −7d vs 0d comparison as both datasets were collected before the vaccination, and so this comparison serves as a control for the other two.

We compare our predictions to those made by Pogorelyy et al., the authors of the original study, for the same six donors [20]. Pogorelyy et al. applied a Bayesian statistical framework to the longitudinal sequence of repertoire snapshots to detect the TCR clones which experienced significant proliferation and contraction, using biological replicates from day 0 to inform a null model of expected proliferation by chance. In contrast to our method, that method uses samples from all of the timepoints. While their predictions do not constitute the ground truth of actual responsive TCR clones to the YFV vaccination, they can still serve as a useful performance benchmark. In particular, we define the "hit rate" as the empirical probability that a clone our procedure detects as responsive was also detected by the original authors as responsive, i.e. the number of sequences we detected as responsive that were in the original responsive set, divided by the number of clones our procedure detects as responsive. We can explore how this "hit rate" varies by timepoint, cluster rank, and donor.

We see that our hit rates behave according to our prior expectations, with larger hit rates for the + 15d comparison, lower but non-neglible hit rates for the + 45d comparison, and virtually no hits for the −7d comparison (Fig 8A). Moreover, the hit rates appear to be generally highest for the top-ranked clusters (i.e., the clusters with the highest loneliness), and decrease to more moderate values as for the lower-ranked clusters (with rank-6 clusters happening to have unusually high rates). We can obtain a smoothed version of these rates by calculating aggregate hit ranks for all clusters up to the given cluster number. For example, when the cluster number is 3, the hit rate is computed over all rank-1, rank-2, and rank-3 clusters. We

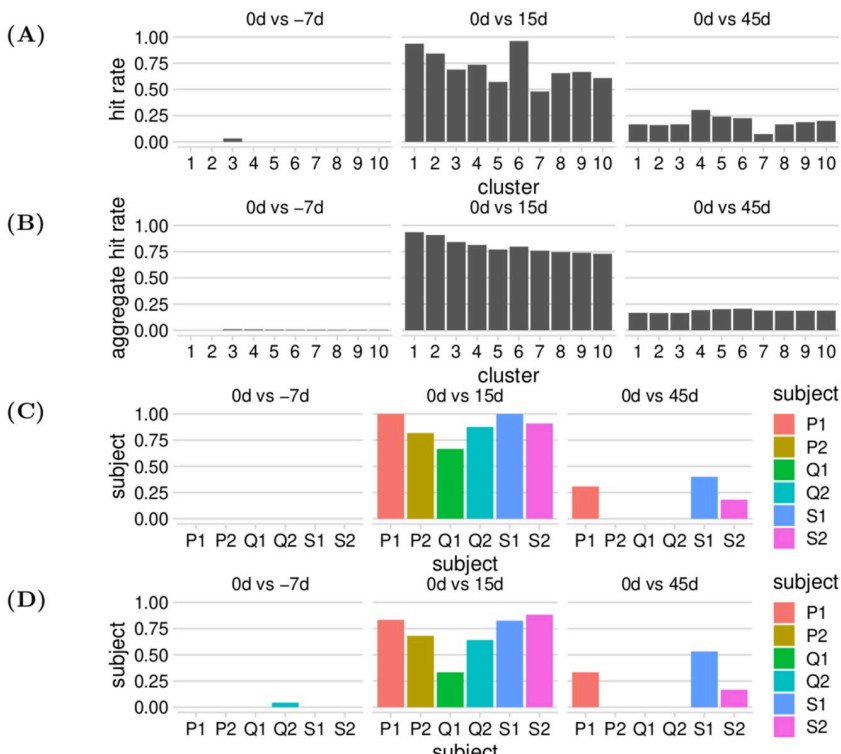

**Fig 8. Various hit rate statistics for the YFV benchmark analysis.** (A) Hit rates of our responsive TCR inferences grouped by reference timepoint and cluster rank. (B) Aggregate hit rates of our responsive TCR inferences grouped by reference timepoint and cluster rank. (C) Hit rates of our responsive TCR inferences grouped by reference timepoint and donor, for cluster rank ≤2. (D) Hit rates of our responsive TCR inferences grouped by reference timepoint and donor, for cluster rank ≤10.

observe a similar pattern, with a steady downward trend for the 15d comparison, and no apparent trend in the other two groups (Fig 8B).

We also see that aggregate hit rates are fairly consistent across subjects, with rates for cluster rank ≤ 2 (i.e., the top two loneliest clusters for each subject) consistently high for the + 15d comparison (Fig 8C), and mostly moderate to high rates for cluster rank ≤ 10 (i.e., the top ten loneliest clusters for each subject) (Fig 8D). Subject Q1 exhibits mildly exceptional behavior, with notably lower hit rates than the other donors in both cases; the original authors also noted some abnormalities for subject Q1 in their analyses, such as comparatively low levels of responsive TCRs on + 15d and + 45d. Moreover, it appears that only subjects P1, S1, and S2 have nontrivial hit rates for both cluster rank ≤2 and cluster rank ≤ 10. These three subjects had the highest + 15d hit rates in general, which suggests that the responsive clusters we found for day 15 were able to persist until day 45, or perhaps suggests a correlation between the strength of the immune response for these two timepoints.

To further assess whether our method is able to detect responsive clusters, we follow the validation of Pogorelyy et al. and examine an independent dataset of public TCRs obtained from VDJdb [45]. This dataset contains 264 sequences of TCRs previously shown to be responsive to a particular YFV epitope, as well as a control set of 370 sequences of TCRs responsive to an unrelated cytomegalovirus (CMV) epitope; call these the YFV validation set and the CMV validation set, respectively. We will say that a TCR for an individual is candidate-responsive if it is in a top-10 cluster (i.e. one of the top 10 highest ranked clusters by loneliness) for the

**Table 1. Counts of matches between our inferred responsive yellow fever (YFV) sequences and either (YFV) or cytomegalovirus (CMV) sequences obtained from VDJdb, where the CMV sequences are used as a control. Also provided are analogous counts for responsive sequences inferred by Pogorelyy et al. [20]. Columns S1—Q2 correspond to the six subjects discussed in [20], also discussed in the Materials and Methods section.**

| Method | Antigen | S1 | S2 | P1 | P2 | Q1 | Q2 | Total |
|---|---|---|---|---|---|---|---|---|
| Ours (exact match) | CMV | 0 | 0 | 0 | 0 | 0 | 1 | 1 |
| Ours (is in a top-10 cluster) | CMV | 2 | 2 | 4 | 4 | 4 | 12 | 28 |
| Ours (exact match) | YFV | 1 | 1 | 1 | 0 | 0 | 0 | 3 |
| Ours (is in a top-10 cluster) | YFV | 28 | 20 | 18 | 11 | 2 | 14 | 93 |
| Pogorelyy (exact match) | CMV | 0 | 0 | 0 | 0 | 0 | 0 | 0 |
| Pogorelyy (1 CDR3aa mismatch) | CMV | 0 | 0 | 0 | 1 | 0 | 2 | 3 |
| Pogorelyy (2 CDR3aa mismatch) | CMV | 5 | 5 | 5 | 3 | 2 | 10 | 30 |
| Pogorelyy (exact match) | YFV | 3 | 5 | 2 | 1 | 3 | 4 | 18 |
| Pogorelyy (1 CDR3aa mismatch) | YFV | 24 | 10 | 12 | 9 | 5 | 21 | 81 |
| Pogorelyy (2 CDR3aa mismatch) | YFV | 27 | 30 | 24 | 11 | 40 | 21 | 153 |

individual's + 15 day comparison. Define the *candidate set* of responsive TCRs to be the union of candidate-responsive TCRs across the six individuals. We use these definitions to compute two quantities for the YFV and CMV validation sets: the number of exact sequence matches found in our candidate set, as well as the number of sequences which belong to any of the top-10 clusters underlying our candidate set. In comparison, Pogorelyy et al. reported the number of exact sequence matches present in their candidate set, the number of sequences with no more than 1 CDR3aa mismatch from some TCR in their set, and the number of sequences with no more than 2 CDR3aa mismatches from a TCR in their set. These two respective comparative methods reflect the way their corresponding inferential methods identify responsive sequences.

We find that our methods are able to identify YFV sequences in the validation set while avoiding CMV sequences in the control set at comparable rates to the methods of Pogorelyy et al. Table 1 shows the results of the above experiment, as well as the results from Pogorelyy et al. (obtained from Table S2 of [20]). Our method detects 3 exact sequence matches to the YFV validation set, and 1 exact sequence match to the CMV validation set. Further, we detect 93 YFV validation sequences and 28 CMV sequences present in our candidate clusters, leading to a true positive/false positive ratio of $93/28 \approx 3.3$. In contrast, Pogorelyy detects a total of 18 exact sequence matches to the YFV set and zero exact matches to the CMV set. When allowing up to 1 CDR3aa mismatch, they detect 81 YFV sequences and 3 CMV sequences, leading to a true positive/false positive ratio of $81/3 = 27$. When allowing up to 2 CDR3aa mismatches, they detect 153 YFV sequences and 30 CMV sequences, leading to a true positive/false positive ratio of $153/30 = 5.1$. While we expect their approach, which uses the full trajectory of datasets across five timepoints for each subject, to perform better in this regard, our approach achieves performance comparable to the Pogorelyy method when allowing two CDR3aa mismatches while only using two timepoints for each subject. Overall, this provides further evidence that our lonely clusters are able to extract YFV-responsive TCR clusters consistently across subjects, and that these clusters generalize beyond the training datasets.

There are a couple of explanations for the discrepancies that do arise between the inferred hits. First, as already mentioned, the set inferred by Pogorelyy et al. is not actually the ground truth of YFV-responsive TCRs, and both methods likely contain false positives as well as true negatives, both of which will impact the hit rates in Fig 8. Further, as mentioned, the approach of Pogorelyy used the full trajectory of repertoire snapshots to infer their set of responsive TCRs, whereas our method only looks at two snapshots at a time. In particular, our inferred

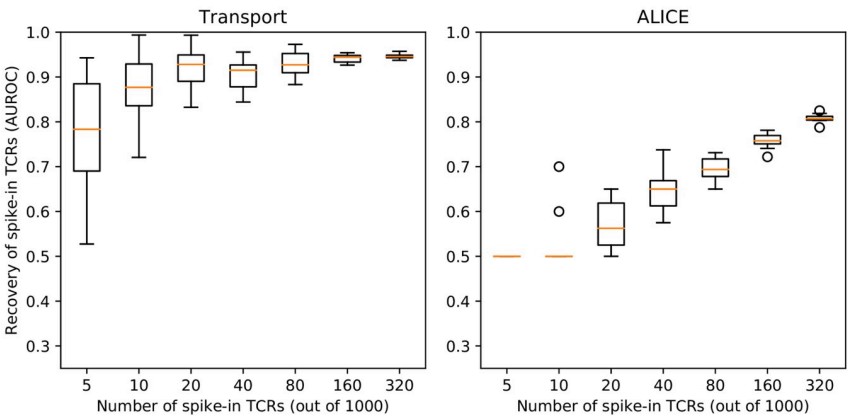

**Fig 9. Recovery of "spike-in" epitope-specific TCRs from background naive TCRs based on optimal transport (left) or the ALICE algorithm (right).** The bars summarize AUROC values for 10 random replicates.

positives corresponding to the + 15d clusters make use of a fraction of the data used by Pogore-lyy, yet we still identify a notable amount of their inferred positives, while avoiding a problematic false positive rate.

## Simulation-based benchmarking

In order to understand the performance of the optimal transport method in a setting where ground truth is known, we performed a simulation experiment using baseline repertoires and "spike-in" sequences that are responsive to a given epitope. Specifically, as background repertoires we used a random sample of 1,000 naive T cell clonotypes from a large data set of $CD8^+$ T cells made available by 10X Genomics [42]. The foreground repertoire consisted of another random sample of naive $CD8^+$ T cell clonotypes along with a variable number of "spike-in" TCR sequences specific for the influenza $M1_{58}$ epitope presented by HLA-A$^*$02:01, such that the entire sample had 1,000 TCRs.

We sought to determine if our NeighborhoodLoneliness value could discriminate the spiked-in sequences from the baseline repertoire. To do so, we treated this as a classification problem and used the Area Under the Receiver Operating Characteristic curve (AUROC), where AUROC values above 0.5 constitute evidence of successful classification, with the evidence strengthening as the AUROC approaches 1. The procedure was repeated 10 times for each number of spike-ins.

We found that our procedure readily distinguishes the spiked-in sequences from the rest (Fig 9), with the AUROC for discrimination of flu-specific TCR sequences increasing as the fraction of spiked-in sequences increases. As a baseline for comparison, we also ran the ALICE program [21] on the same data. ALICE uses a parametric background model to identify enriched TCR neighborhoods present in a single TCR repertoire. ALICE was run on the foreground TCRs, with default parameter settings and 100 million simulated sequences, and an AUROC value was calculated by sorting the TCRs based on the ALICE p-value. ALICE also succeeded in discriminating the spike-in TCRs from the background naive TCRs, though with lower AUROC values than the optimal transport measure.

## Discussion

We have described a nonparametric approach to TCR repertoire comparison driven by optimal transport and TCRdist, including a novel clustering algorithm that determines differential

enrichment of collections of TCRs between two repertoires. We demonstrated that our framework can successfully extract biologically meaningful differences between distinct TCR populations through several analyses. Our methods were able to identify several clusters that are consistently enriched in the double negative T cell repertoire with respect to the CD4$^+$ T cell repertoire across biological replicates, and characterize their V gene and CDR3aa profiles. These clusters have significant overlap with clusters that were hand-identified by independent and close examination of a TCR data set. We also presented a randomization test to obtain significance estimates of our TCR scores, and validated them against a proxy null distribution comprised of the double negative biological replicates. Finally, our methods were able to detect responsive TCR clusters to a yellow fever virus immunization across multiple donors using only one post-vaccination repertoire snapshot per donor.

Our framework can be viewed as a nonparametric approach to detecting collections of TCRs in a target repertoire that are enriched compared to a specified null distribution, which is manifest as a source or reference repertoire. Thus, the inferences will be valid insofar as our source repertoire is a representative sample from the underlying population of interest. This provides flexibility in which baseline distribution to compare against if we do have a reference repertoire that we are confident represents the population of interest, rather than relying on a model that might be biased towards a different or more general population. Furthermore, our method fundamentally involves comparing collections of amino acid sequences via a distance rather than exact identity, making it more robust to sequencing error than methods based on exact matches of nucleotide sequence.

The choice of whether to use a model-based method versus our non-parametric method is analogous to the choice between a t-test and a nonparametric test such as the Mann-Whitney U. If we do not have a representative sample repertoire from the population of interest, a model-based approach might yield more robust results. On the other hand, if data is plentiful and we are concerned about model fit, a nonparametric test may be more appropriate. Thus, one must decide which reference population should be used for the particular application, and how this reference population can be best represented, in order to choose the appropriate approach.

There are several avenues to improve and extend the present work. We note that our current clustering mechanism (Algorithm 1) does not automatically estimate the number of clusters to return. Both of these considerations are in contrast to parametric approaches like ALICE [21], which rely on a parametric $P_{gen}$ model for the null distribution but can be directly applied to a single repertoire and automatically return any significant cluster. Though we have focused on single-chain repertoire data in this study, the TCRdist measure applies equally well to paired alpha/beta TCR sequences, and the optimal transport framework presented here should be well suited for analysis of paired repertoires from single-cell genomics experiments. In this context, it may be interesting to compare and constrast optimal transport repertoire analysis with parallel analyses of the transcriptional phenotypes of the same T cell clonotypes.

Another future direction involves trying other distance functions between immune receptors, and seeing how other metrics impact the results. This could also lead to a generalization of our methods to B cell receptor (BCR) repertoire data, as there is no current equivalent to TCRdist for BCRs. One possible direction would be to examine the efficacy of BCR and TCR sequence embeddings within our optimal transport framework, such as the embeddings underlying recent variational autoencoders for TCR sequences [46].

One might also try another distance between probability distributions that also incorporates a metric function on the individual objects in the sample space. Perhaps the two most popular alternative distances between two probability distributions are known as the discrepancy metric and the Prokhorov metric. The discrepancy metric between probability measures

$\mu$ and $v$ is defined as

$$d_{\text{Discrepancy}}(\mu, v) := \sup_{\text{closed balls } B} |\mu(B) - v(B)|. \tag{23}$$

In other words, this metric looks at every possible ball, calculates the absolute difference in probability measures of the ball, and gives the largest such difference. Such a metric is oblivious to other differences occurring in the region space, and thus, seems less appropriate for distributions over TCRs where there could be many sub-distributions with interesting behavior. The Prokhorov metric between $\mu$ and $v$ is defined as

$$d_{\text{Prokhorov}}(\mu, v) := \inf\{\varepsilon > 0 : \mu(B) \leq v(B^\varepsilon) \ \forall \ \text{closed balls } B\}, \tag{24}$$

where $B^\varepsilon = \{x : \inf_{y \in B} d(x, y) \leq \varepsilon\}$. Similarly to the discrepancy metric, this metric fixates on an infimum over the region and fails to account for more subtle differences between distributions. Thus we believe that the optimal transport metric is the most appropriate in the TCR setting.

## Supporting information

**S1 Fig. Neighborhood loneliness is a better discriminator of "spike-in" TCRs than individual (per-TCR) loneliness.** Each bar summarizes the Area Under the Receiver Operating Characteristic curve (AUROC) values for 10 replicate experiments in which a varying number (x-axis) of TCRs sharing a single epitope specificity were spiked into one of two repertoires randomly sampled from a large population of naive CD8+ T cells. The AUROC measures the ability of the corresponding measure (neighborhood loneliness on the left or individual loneliness on the right) to differentiate the spiked-in TCRs from the naive CD8+ TCRs when comparing the two repertoires.
(TIFF)

**S2 Fig. Scatterplots of mean annulus loneliness vs TCRdist radius for each of the DN repertoires, along with estimated segmented regression fits. Repertories with fewer than 200 TCRs have a dashed regression line.** This is a visual check of the assumptions of the segmented regression specified by Eq (14), showing that the assumptions hold in the relationship of mean annulus loneliness versus TCRdist from the centroid, and that this relationship has identifiable breakpoints for a typical repertoire. We perform Algorithm 1 on each full repertoire, which yields the "loneliest" cluster of each repertoire. We see that we are able to successfully estimate a breakpoint $r_{\text{breakpoint}}$ for each subject, with $r_{\text{breakpoint}} \in (50, 100)$ for almost all subjects. When the repertoire contains fewer than 200 TCRs, the relationships can weaken (e.g. Subject 1), though Algorithm 1 still provides sensible regression estimates. When the repertoire contains at least 200 TCRs, we see consistent elbow behavior and convincing breakpoint estimates. Furthermore, violations of the least squares assumptions do not appear to be a concern.
(TIFF)

**S3 Fig. Time breakdown for a collection of pairwise analyses, comparing the amount of time required for the TCRdist calculation to the time required for the optimal transport (OT) calculation.**
(TIFF)

## Acknowledgments

The authors would like to thank Lucy Yang and Jared Galloway for their work trying out and improving the software described in this manuscript.

## Author Contributions

**Conceptualization:** Branden J. Olson, Paul G. Thomas, Philip Bradley, Frederick A. Matsen IV.

**Data curation:** Branden J. Olson, Stefan A. Schattgen, Paul G. Thomas.

**Formal analysis:** Branden J. Olson, Frederick A. Matsen IV.

**Funding acquisition:** Paul G. Thomas, Philip Bradley, Frederick A. Matsen IV.

**Investigation:** Branden J. Olson, Stefan A. Schattgen, Paul G. Thomas.

**Methodology:** Branden J. Olson, Philip Bradley, Frederick A. Matsen IV.

**Project administration:** Philip Bradley, Frederick A. Matsen IV.

**Resources:** Paul G. Thomas.

**Software:** Branden J. Olson, Philip Bradley, Frederick A. Matsen IV.

**Supervision:** Philip Bradley, Frederick A. Matsen IV.

**Validation:** Branden J. Olson, Stefan A. Schattgen, Philip Bradley.

**Visualization:** Branden J. Olson, Philip Bradley.

**Writing – original draft:** Branden J. Olson, Philip Bradley, Frederick A. Matsen IV.

**Writing – review & editing:** Branden J. Olson, Stefan A. Schattgen, Paul G. Thomas, Philip Bradley, Frederick A. Matsen IV.

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
