## [Decision Letter · Decision Letter 0]

26 Apr 2022

Dear Dr. Matsen IV,

Thank you very much for submitting your manuscript "Comparing T cell receptor repertoires using optimal transport" for consideration at PLOS Computational Biology.

As with all papers reviewed by the journal, your manuscript was reviewed by members of the editorial board and by several independent reviewers. In light of the reviews (below this email), we would like to invite the resubmission of a significantly-revised version that takes into account the reviewers' comments.

Your revised manuscript will be sent to the reviewers for further evaluation. They raised several substantial concerns; please address these as carefully as possible, and note that we cannot guarantee acceptance of the revised version.

Sincerely,

Andrew J. Yates

Associate Editor

PLOS Computational Biology

Rob De Boer

Deputy Editor

PLOS Computational Biology

Reviewer's Responses to Questions

**Comments to the Authors:**

Reviewer #1: Summary

Olson and colleagues present a method for comparing TCR repertoires and detect significant differences between them. Comparison of two repertoires is formulated in terms of a classical computer science problem (discrete optimal transport) with a TCR-specific metric (TCRdist). The authors provide a concise formal definition of this problem and elegantly adapt it to TCR repertoires. They also describe a statistical procedure for testing if the calculated repertoire difference is significant and show how to identify TCRs and motifs that are responsible for the difference. The authors validate this statistical procedure with experimental data from biological replicates and apply their method to longitudinal data from individuals vaccinated against yellow fever to identify post-vaccination shifts in the TCR repertoires.

However, the authors did not test their method on ground truth data (simulated) rendering an evaluation of the method difficult. More generally, the figures and test are nearly impossible to understand, Therefore, the real-life application of the method is unclear. The major and minor issues related to paper are discussed below:

Major issues

- Biological usefulness: the most biological claim is that “the framework can successfully extract biologically meaningful regions between distinct TCR populations” but the meaning of these regions is unclear.

- TCRdist is known in the community for high running time. In the current manuscript, the authors avoided running time problems by analyzing relatively small samples: “Each repertoire is filtered to the 1,000 most abundant clones”. The robustness of the method to such downsampling needs to be shown, and running time statistics for the computations in the manuscript need to be provided. Specifically, it would be great to see simulations to understand the sensitivity of the method.

- AIRR analyses often tend to be sensitive to data preprocessing (clonotype computation etc.). Does the described method require for both repertoires under comparison to be preprocessed in an identical way? Will the comparison conclusion hold if both datasets are first processed identically in one way, and then identically another way (e.g. with different preprocessing tool parameters). Generally, robustness of the described method to preprocessing differences needs to be shown or at least explained.

- The manuscript lacks comparison (at least a discussion thereof) of the described method with other methods for detecting repertoire difference (and even citation of them) beside Pogorelyy et al. PNAS 2018: for example, Dupic et al. PLOS Genetics 2021 (https://pubmed.ncbi.nlm.nih.gov/33395405/), Weber et al. bioRxiv 2022 (https://www.biorxiv.org/content/10.1101/2022.01.23.476436v1.full), Mayer-Blackwell elife 2021 (https://elifesciences.org/articles/68605), Slabodkin et al. Genome Research 2021 (https://genome.cshlp.org/content/early/2021/11/23/gr.275373.121.abstract), Alon Front Imm 2021 (https://www.ncbi.nlm.nih.gov/pmc/articles/PMC8047331/). Repertoire comparison is *the* main challenge of AIRR analysis. Please cite the literature appropriately.

- The developed statistical test requires validation: the authors should show the p-value distribution in the case when the null hypothesis is true. And again, simulations would be nice as they would allow stress-testing the method.

“We wished to develop a procedure that performs comparisons between two empirical repertoires in a fast, interpretable, and precise manner” → where is the manuscript do you explain “interpretability”? The plots in the results section are overall very hard to understand (as is the text, please streamline), so it is not clear to us how this method can be practically used for repertoire comparison. We might also have a different definition of “comparison”. It seems to us that your method is to point enriched clones versus a baseline? One can call this comparison, but one could have also been more direct as for example in this paper’s title https://elifesciences.org/articles/68605.

- “While their predictions do not constitute the ground truth of actual responsive TCR clones to the YFV vaccination, they can still serve as a useful performance benchmark” → why is another method a useful performance benchmark? And if the method by Pogorelyy is so useful as a benchmark, in what why is your method novel (or can bring about new biological insight which was not possible with Pogorelyy's method)?

Minor issues

- The equations are denoted the same as references. Using “Eq.” prefix for all equation references will make the text easier to read.

- Why is step size for clustering set to 5 by default?

- The “Soldiers on a battlefield” example needs to be formalized a bit further: it is not entirely clear from the description whether we consider all soldiers different or not. More generally, do we really need a war analogy in scientific papers on TCRs?

- Code availability: it is said that all the code is available on Github but there is no link in the manuscript

- The more focused problem of inferring specificity from TCR sequences has been approached by tracking individual clones, comparing to a probabilistic model, or using epitope-specific machine-learning models. → please provide citations for each claim.

- Other machine learning techniques build predictive models using labeled training data [16–18], although these techniques often require a specified antigen epitope, can be limited by the amount of publicly-available data, and rely on models that can be difficult to interpret. this sentence sounds like your approach will also be able to compare epitope-specific repertoires. Please rephrase.

- For the datasets used, please mention how preprocessing was performed.

Please define biological replicate.

Reviewer #2: Adaptive immune recognition relies on an incredibly diverse set of transmembrane receptors diversified through genetic recombination. The ability to read out this diversity through sequencing allows measurement of this diversity at unprecedented scale. To make good on the promise of repertoire sequencing to provide new insights into adaptive immunity, there is an important need for better statistical and computational analysis techniques for this complex data. In the paper 'Comparing T cell receptor repertoires using optimal transport', the authors propose using the mathematical framework of optimal transport as an elegant way of comparing similarity between T cell receptor repertoires. By building on recent advances in the field of optimal transport, namely the Sinkhorn distance formalism, the authors demonstrate computational tractability of the approach. Importantly, the paper also provides some evidence that their approach can detect biologically meaningful differences between samples in case studies.

The proposal is conceptually innovative and the paper is well-written overall. However, as currently presented there are a number of concerns regarding the statistical foundations of the method and its benchmarking that should be suitably addressed.

Major concerns/comments/question:

- Can the definition of the relative loneliness measure be given a less heuristic motivation? Or alternatively, can the consequences of this definition be better explored on a toy model? Are there any insights from theory into how to choose the neighborhood size delta?

- The fit in Fig.6 has a slope significantly below one, which implies that the randomization z-scores consistently overestimate significance. This deserves explanation. Note that in applications where biological replicates are not available this severely limits the practical utility of the method.

- In the last results section, a more fair comparison would be with 1 CDR3aa mismatch as the number of true positives using this threshold is closer between both methods. The true positive to false positive ratio for the benchmark method is then 81/3 = 27. This implies a very different conclusion regarding the relative performance of the methods.

- An important comparison to the ALICE method is missing from the current manuscript. Such a benchmarking is important as it is more direct than with the longitudinally identified sequences, as discussed by the authors. How well does the non-parametric method perform relative to ALICE, which uses additional information from its learned parametric model of recombination? An inferior performance might still make the simpler method presented here useful, but it is important to have an idea of how much statistical power is lost.

Minor concerns/comments/questions:

- The results only apply the loneliness measure to data, but in certain contexts the overall optimal transport distance might be interesting in its own right and could be illustrated in an application. A comparison and/or discussion of the optimal transport distance with distance measures that might be constructed from sequence-similarity weighted repertoire diversity measures might also improve the manuscript.

- What is the biological motivation for analyzing outliers in DN with respect to CD4 in the first results section? Intuitively, the reverse comparison seems more biologically meaningful given the developmental lineage of T cells.

- On line 125, it might help readability to define D in terms of x_i, y_j, as D_ij = d(x_i, y_j).

- The definition of the candidate set of TCRs and all sequences in any of the top10 clusters on lines 578-583 should be clarified. It remains unclear to me what precisely is meant by each.

- The value of delta used in the results section should be indicated in the legend/text.

- A link to the github repository mentioned in the data/code availability statement should be added.

Typos:

line 159: abT -> rcT

line 274: to *be* estimated

Reviewer #3: This study introduces a clever strategy to compare and analyze TCR repertoire distributions using the optimal transport method. The advantage that this strategy offers over the probability generation models that identify unique clusters in a given repertoire is that it uses a combination of probability mass distribution and the similarity-matched distance metric to identify uniquely enriched TCR sequences in a repertoire under consideration as compared to a reference repertoire. The study provides sufficient explanation of the working principle of the method and the evidence of its applicability to publicly available datasets. The independence from modelling assumptions and parametric approximations can be viewed as the strength, however the success of this approach heavily relies on the quality of the data and availability of a compatible reference repertoire. It may be helpful if authors highlight the unique insights that may emerge from using this method to compare TCR distributions (e.g. see major point 2 below), in addition to focussing on benchmarking its performance in comparison to other studies.

Major points:

1. The major concern using a reference repertoire to gain insights about a test repertoire is context dependency. For example, it is possible that a clone responding to a vaccination strategy is also enriched in the pre-vaccination reference repertoire due to either high probability of generation (and peripheral selection) or due to prior immunization experience. Such clone would not be picked up as "lonely" using this approach. Conversely, a relatively weakly responding clone may have a very low abundance in the reference repertoire and thus would rank high on the lonely scale. How does this approach handles such clonal abundance disparity?

2. This approach may also allow for the analysis and quantification of inter-individual variability in the immune responses. For example, the authors could compare post vaccination d15 TCR repertoires of individuals immunized with yellow fever vaccine, which may reveal valuable insights about the differences in clonal distributions and how they affect the dynamics of response to the vaccine. Especially, in individuals with high and low hit rates between 0d and 15d comparisons (P1 and Q1, for example).

3. The definitions of w(c) and AAdist need justification (Lines 183 and 184).

4. Why do authors say that predictions from Pogorelyy et al. do not constitute the ground truth of actual responsive TCR clones to YFV vaccination. Please justify respectfully. (Line 545).

5. During clustering, does the Algorithm 1 reach the breakpoint because no more sequences are found in increasing radii or because too many sequences are added to the cluster such that the effective decrease in the mean loneliness is substantially small?

Minor points:

1. Discussion of the productivity-based filters that limit the diversity of circulating pool of TCR sequences is needed. (Line 9-10).

2. Equation numbering needs to be careful and consistent. Some equations are referred in the text as Eq. XX while some are just referred (XX). Also line 452 should be Eq. 18.

3. Define “hit rate” properly (line 549).

**Have the authors made all data and (if applicable) computational code underlying the findings in their manuscript fully available?**

Reviewer #1: **No: **Github code is not mentioned in the manuscript.

Reviewer #2: None

Reviewer #3: None

PLOS authors have the option to publish the peer review history of their article (what does this mean?). If published, this will include your full peer review and any attached files.

Reviewer #1: No

Reviewer #2: No

Reviewer #3: **Yes: **Sanket Rane
---

## [Decision Letter · Decision Letter 1]

24 Oct 2022

Dear Dr. Matsen IV,

We are pleased to inform you that your manuscript 'Comparing T cell receptor repertoires using optimal transport' has been provisionally accepted for publication in PLOS Computational Biology.

Best regards,

Andrew J. Yates

Academic Editor

PLOS Computational Biology

Rob De Boer

Section Editor

PLOS Computational Biology

Reviewer's Responses to Questions

**Comments to the Authors:**

Reviewer #1: The authors have addressed all my comments.

Reviewer #2: The authors have suitably addressed the questions raised during review. This has improved the manuscript, which in my view can be published without further revision.

Reviewer #3: Satisfied with the revised version and authors' response. No further comments.

**Have the authors made all data and (if applicable) computational code underlying the findings in their manuscript fully available?**

Reviewer #1: Yes

Reviewer #2: Yes

Reviewer #3: Yes

PLOS authors have the option to publish the peer review history of their article (what does this mean?). If published, this will include your full peer review and any attached files.

Reviewer #1: No

Reviewer #2: **Yes: **Andreas Mayer

Reviewer #3: **Yes: **Sanket Rane

---

## [Editor Report · Acceptance letter]

11 Nov 2022

PCOMPBIOL-D-22-00401R1 

Comparing T cell receptor repertoires using optimal transport

Dear Dr Matsen IV,

I am pleased to inform you that your manuscript has been formally accepted for publication in PLOS Computational Biology. Your manuscript is now with our production department and you will be notified of the publication date in due course.

With kind regards,

Zsofi Zombor
